# OpenReview forum: "Black-Box Adversarial Attacks on LLM-Based Code Completion"
_ICLR.cc/2025/Conference — Submitted to ICLR 2025_

### Official Review · Reviewer_Ue9Z · 2024-10-28

**Soundness:** 3
**Presentation:** 3
**Contribution:** 2
**Rating:** 5
**Confidence:** 3

**Summary:**

The paper introduce INSEC, a novel black-box adversarial attack designed to manipulate LLM- based code completion engines into generating insecure code by inserting attack strings as comments in code completion input. INSEC first applies five different approaches to generate initial attack strings, then optimizes them through random mutations. Through iterations of mutation and selection, INSEC maintains a pool of most effective attack strings, and tests their effectiveness on different large language models. The authors propose a dataset containing 16 CWE and 5 programming languages, and evaluate INSEC on open-source models and commercial services. Results show a 50% rise in insecure code generation without majorly affecting functional correctness. The method is also cost-efficient, requiring less than $10 for querying APIs, and doesn’t need access to model internals.

**Strengths:**

1. The paper presentation is quite good, easy to follow. The authors first introduce the background information of code completion and threat model, then detailed explain the implementation of INSEC step by step, and finally present an extensive evaluation.

3. The experiments are rich and involve a large workload. The authors analyze many hyperparameters, including the insertion position of attack strings and the impact of different tokenizers on effectiveness of attack, making the result solid.

4. The paper propose the first black-box adversarial attack on LLM-based code completion system, and introduce a novel method for initializing attack strings, with five well- designed methods and analysis.

**Weaknesses:**

1. Technical wise, the paper appears rather shallow. This is my main concern. The mutate method seems simple, with only random functions to decide mutation position and replacement. Despite that applying random mutate to optimize the initial attack string could construct effective attack strings, it may not be efficient and could lead to local optima.

2. Evaluation can be enhanced. There is no comparative analysis of different mutate methods and experimental analysis of the mutation process. I notice that you implemented many optimization methods in the submitted code, but there is no experiment results comparing them, Why?

3. Models. The experiments didn't include some SOTA models like GPT-4. More SOTA models should be included for persuasiveness.

4. (stealthiness) No discussion/evaluations on defense methods. For instance, would the mutated inputs (e.g., those "dalЖ +k重d5" in Figure 1) be easily detected by some anomaly detection tools?  Discussions on stealthiness mostly on the "output functionality"; this appears rather thin.

**Questions:**

1. In page 8, Figure 5, there are a column of Opt only, I am wondering how do you form a set of attack strings with only optimization method? There should be initial string for optimization.

2. I notice that you implemented many optimization methods in the submitted code, but there is no experiment results comparing them, Why?

3. What is the result of INSEC on GPT 4?

4. Can some anomaly detection tools easily detect the attack? In other words, how to understand the stealthiness of the attack, from the input perspective?

---

> ### Author Response · Authors · 2024-11-25
> **Response to Reviewer Ue9z**
>
> We would like to thank the reviewer for their insightful review and constructive comments. We address their comments, questions, and concerns below:
>
> ### **Q1: Does the simplicity of the optimization procedure of INSEC limit its practical relevance as an attack on the security of code completion engines?**
>
> No. In fact, we believe it increases the severity, since it makes the attack more accessible and easier for real-world attackers to execute.
>
> ### **Q2: Could you provide results on other models?**
>
>
> The INSEC attack is designed for Fill-in-The-Middle completion—the format that is used in IDE-integrated code assistants. We evaluated INSEC against the state-of-the-art and industry standard in this domain, Copilot, and successfully broke its self-claimed safeguards to actively prevent vulnerable completions [1]. We evaluate other open-weight state-of-the-art completion models and the latest completion-API compatible model by OpenAI, gpt-3.5-turbo-instruct. Other recent models, such as the suggested GPT-4, are chat models, and not suitable for code completion and INSEC.
>
> ### **Q3: Could anomaly detection tools detect the attacked inputs?**
>
> We believe that the attack is quite stealthy. The injection affects only the prompt to the language model, which would need to be monitored by such a tool. Further, the assumption that such detection tools are present in production setups may be unrealistic, since they are difficult to set up and expensive to maintain [2]. We have added a general discussion on defenses in Appendix E of the paper and refer to Q2 of reviewer Jqc8 for additional discussion of possible defenses.
>
> ### **Q4: How do you form the attack string in the “optimization only” setting?**
>
> We utilize the random initialization strategy for 5 attack strings in the optimization-only setting. We have added this detail in Section 5.3.
>
> ### **Q5: Why are there multiple optimization methods implemented in the code but no comparison among them in the paper?**
>
> Most of those implemented methods are based on white-box access and were removed from the final version of the paper, which focuses on black-box attacks. The only alternative mutation method differs by deciding to mutate characters individually per character, resulting in an exponentially decreasing probability to mutate several characters, compared to INSEC, which samples the amount of characters to change uniformly. We found that these methods perform overall similarly, confirming that INSEC is a generic and effective attack. Most importantly, and adding to the Q1, both methods are realistic choices for attackers, as they are simple to implement. We summarize our findings in Appendix C.
>
> **References**
> [1] https://github.blog/ai-and-ml/github-copilot/github-copilot-now-has-a-better-ai-model-and-new-capabilities/#filtering-out-security-vulnerabilities-with-a-new-ai-system
> [2] Ita Ryan, Utz Roedig, and Klaas-Jan Stol. 2023. Unhelpful Assumptions in Software Security Research. CCS 2023

---

> ### Author Response · Authors · 2024-12-02
>
> We would like to kindly remind the reviewer that the discussion period is approaching its end this Tuesday. We believe to have provided an extensive rebuttal to the points and questions raised by the reviewer. In case the reviewer has any remaining concerns or questions that support their leaning towards rejection, we are eager to provide further clarifications.

---

### Official Review · Reviewer_Jqc8 · 2024-11-02

**Soundness:** 3
**Presentation:** 3
**Contribution:** 2
**Rating:** 5
**Confidence:** 4

**Summary:**

This paper introduces INSEC, a black-box adversarial attack targeting large language models (LLMs) for code completion, specifically aiming to manipulate these models into generating vulnerable code. The attack operates by injecting a crafted comment as a prompt, subtly influencing the model's output toward insecure code. Evaluations across multiple state-of-the-art models, including open-source engines and black-box commercial APIs like GitHub Copilot and GPT-3.5-Turbo-Instruct show that INSEC can increase the vulnerability rate of code generated while maintaining functional correctness, posing security risks for deployed LLM-based code assistants.

**Strengths:**

- This paper proposes INSEC, a new black-box adversarial attack to real-world code completion systems.
- Extensive experiments include vulnerability and functional correctness metrics, demonstrating INSEC's efficacy across seceral LLM-based engines.

**Weaknesses:**

- The practicality of the proposed threat model for code LLMs could be further clarified.

First, while the adversarial attack demonstrates a creative approach, its real-world applicability may be limited. This is because the attack relies on embedding a specifically crafted comment adjacent to the exact code snippet that the user wants completed. In real scenarios, users will not attack the code LLM to generate vulnerable code for themselves. Even if an attacker were to attempt this, the suggested deployment methods in the paper, such as malicious IDE plug-ins or other input control techniques, may face practical challenges. In dynamic or interactive environments, where code is generated line by line, integrating a crafted comment seamlessly within existing code snippets might be difficult. This differs from prompt-response interactions typical of natural language LLMs, where such manipulations might be simpler to execute covertly.

Additionally, even if the attack succeeds in generating vulnerable code, static analysis tools commonly used before code deployment may readily detect these vulnerabilities. Given that code undergoes security scanning, this could limit the attack's effectiveness.

- Selection and Mutation Strategy.

The approach of using selection and mutation strategies, while effective, is not entirely novel for adversarial attacks on code LLMs. Similar techniques have been applied in other adversarial attack settings, such as [1]. Additionally, while CodeQL was selected as the primary vulnerability detection tool, not all static analysis tools cover the full spectrum of CWEs. It may be worth considering alternative or additional tools to provide a broader assessment of the attack's impact.

- It appears that no defense strategies are discussed in the paper.

[1] Yang, Z., Shi, J., He, J. and Lo, D., 2022, May. Natural attack for pre-trained models of code. In Proceedings of the 44th International Conference on Software Engineering (pp. 1482-1493).

**Questions:**

- In Figure 2 and Figure 3(a), the first part of each figure appears to be under the same settings, yet the results show slight differences (74 vs. 73, and 78 vs. 77). Could it be clarified if there are any specific reasons behind these discrepancies?
- In the “without comment” setting, it appears that the crafted comment is inserted without the comment symbol. My concern is that without this symbol, the inserted string could interfere with the generated code's functionality by introducing compilation errors. This might make the setting less realistic or impactful, and I would appreciate any clarification on how this potential issue is addressed.
- Have the paper investigated whether the crafted comment has transferability? Specifically, could a comment crafted for one model effectively trigger vulnerabilities in other models?
- Are there any discussions on defense mechanisms against this attack? I imagine that standard static analysis tools or even simpler measures, like filtering out all comments, could mitigate this attack effectively. Additionally, have the paper considered adaptive attacks that might bypass straightforward defenses?

---

> ### Author Response · Authors · 2024-11-25
> **Response to Reviewer Jqc8 (Part 1)**
>
> We would like to thank the reviewer for their insightful review and constructive comments. We address their comments, questions, and concerns below:
>
> ### **Q1: Can such an attack be executed in practice?**
>
> Yes, as detailed in the paper, we believe that INSEC is highly practical.
>
> First, next to typo-squatting and offering a free completion service to users, practically any extension installed in VSCode may hijack other extensions due to the ability to execute arbitrary commands on the user machine [5]. This may be exploited to directly manipulate the installed, official, Copilot extension from a benign-looking extension offered by the adversary, such as a custom theme. Moreover, malicious Python packages, as recently discovered [6], could pose a similar attack vector.
>
> Second, integration of the attack into generated code is trivial, since we always inject a comment line before the currently edited line by the user. We want to point out here again that we do not require any detection of a good insertion position, since the attack is designed to allow injection at any position.
>
> Finally, users would not notice injections into intercepted prompts, since they obtain functionally correct code from SOTA fill-in-the-middle models, such as Copilot.
>
>
> ### **Q2: Would security analysis before deployment catch the introduced vulnerabilities?**
>
> Yes, but this is not realistic in a general case. First, we would like to point out that a significant part of our dataset is sourced from actual vulnerabilities found on GitHub, as such vulnerabilities are definitely being deployed into production settings. Previous work also reports similar findings [7]. Moreover, analyzer services for code security are difficult to set up, expensive to maintain, and developers are constrained by work environments among other obstacles to successfully deploy security analysis in production settings. Previous work has shown that assuming the presence of analysis services is generally unhelpful in determining real-world threats [8].
>
> ### **Q3: Is INSEC different from [1] and provides significant contributions beyond it even though both use variations of genetic algorithms for optimization?**
>
> Yes, we highlight two key contributions of INSEC over [1] below. Note that the use of genetic optimization does not limit novelty, because it is a general optimization technique applied across many domains leading to novel results (e.g., [2,3,4]). That being said, we thank the reviewer for pointing us to [1] and we have included a discussion of [1] in our revised paper.
> - *Classification vs. Generation*: [1] focuses on classification tasks with fairly outdated BERT models, such as vulnerability detection and authorship attribution. In contrast, INSEC targets code completion with state-of-the-art LLMs, an actual practical and commercial use case with user bases in the millions. As such, our work raises timely practical concerns for critical masses of developers.
> - *Inference-time vs. Training-time Optimization*:  In [1], the attack optimization algorithm has to be performed for each input, which incurs significant overhead during inference time. On the contrary, for INSEC, the short attack string is derived at training time and is reused across all inputs at inference time. This approach is realistic and meets the real-time requirements of modern code completion.
>
> ### **Q4: Does the simplicity of the optimization procedure of INSEC limit its practical relevance as an attack on the security of code completion engines?**
>
> No. In fact, we believe it increases the severity, since it makes the attack more accessible and easier for real-world attackers to execute.
>
> ### **Q5: Is the evaluation of INSEC’s impact limited by using CodeQL to detect vulnerabilities?**
>
> No, since our method does not depend on CodeQL as a CWE detection method. It was chosen due to its customizability, and as it is adopted as standard in prior work [7,9]. Attackers may build their attacks using other static analysis tools, however, the tool and its capabilities are not relevant to the attack itself. In fact, we only utilize specific queries from the extensive repository of CodeQL. Attackers could similarly hand-craft a specialized tool to detect precisely the vulnerabilities that they want to inject, use it for training, and then manually assess the quality of the results. This would even increase the severity of the resulting attack, since the injected vulnerabilities would likely not be detected by potentially deployed, publicly available countermeasures.
>
> We further manually verify the results of CodeQL, by investigating 50 generated completions of the test set from unattacked, Init-only attacked and optimized attacked models, across all CWEs and models. The precision of CodeQL among these samples was 98%. We added the details of our manual code review and a clarification to the paper in lines 316 and 317 and Appendix A.

---

> ### Author Response · Authors · 2024-11-25
> **Response to Reviewer Jqc8 (Part 2)**
>
> ### **Q6: Why are the vul_ratio and func_rate@1 for StarCoder 3B different in Fig 2 and 3a?**
>
> Because Figure 3a is an ablation setting, we evaluate it on the validation split of our vulnerability and functional correctness datasets. In Figure 2, the evaluation results on the test split are presented.
>
> ### **Q7: Does the “without comment” setting impact code correctness?**
>
> No. The code with injected tokens is only used for generating the output of the model, the injection is not present in any compiled or executed code.
>
> ### **Q8: Did you investigate generalizability across models?**
>
> We thank the reviewer for this interesting question. We investigate this by applying the strings optimized for StarCoder-3b and CodeLlama-7b and use them when attacking GPT-3.5. The attack generalizes between models, increasing the vulnerability ratio of GPT-3.5 to 55% and 59%, respectively. Meanwhile, targeted optimization outperforms the transferred attack on both the original models StarCoder-3B and CodeLlama-7B (80% and 82% respectively) and the transferred model GPT-3.5 (76%). We included these findings in Appendix C of the paper.
>
> ### **Q9: What would be possible defense strategies? Did you consider adaptive attacks to these?**
>
> We add a discussion on possible defenses in Appendix E, including scrubbing of comments and deploying static analysis in production environments. In the case of removing comments from code prompts, we suspect that such an invasive change could heavily impact functional correctness, as developers often annotate code with comments as instructions for LLM completion. We confirm this experimentally, by removing all comments from the HumanEval dataset before submitting them to the LLM for completion and observe a func_rate@1 of only 89.6% on average compared to the commented code, resulting in a similar impact to INSEC. Regarding static analysis, we find that, while effective, such tools are generally difficult and expensive to set up and maintain, while developers are often constrained by their environment and budget [8].
>
> We did not explicitly consider adaptive attacks in the presented paper. One can imagine however that due to the general design of our approach, the detectability of model-generated completions by a static analysis tool could be included as a factor for the selection step. To investigate this further, a practical defense should be developed first.
>
>
> **References**
>
> [1] Z Yang et al., "Natural Attack for Pre-trained Models of Code". ICSE 2022.
> [2] T Liu et al., "Generating Private Synthetic Data with Genetic Algorithms". ICML 2023.
> [3] M Z Nawaz et al., "Proof Searching in HOL4 with Genetic Algorithm". SAC 2020.
> [4] S Polu et al., "Formal Mathematics Statement Curriculum Learning". ICLR 2023.
> [5]  Kevin Ward and Fabian Kammel, "Abusing VSCode: From Malicious Extensions to Stolen Credentials (Part 1)". https://control-plane.io/posts/abusing-vscode-from-malicious-extensions-to-stolen-credentials-part-1/
> [6] Leonid Bezvershenko, Tweet: https://x.com/bzvr_/status/1859233190449213880
> [7] Jingxuan He and Martin Vechev. “Large language models for code: Security hardening and adversarial testing”. CCS, 2023
> [8] Ita Ryan, Utz Roedig, and Klaas-Jan Stol. 2023. "Unhelpful Assumptions in Software Security Research". CCS 2023
> [9] Hammond Pearce et. al. “Asleep at the Keyboard? Assessing the Security of Github Copilot’s Code Contributions”. IEEE S&P, 2022

---

> > ### Comment · Reviewer_Jqc8 · 2024-11-26
> >
> > Thank you for your response. While it addresses some of my concerns, I still have reservations.
> >
> > Firstly, I remain unconvinced about the practicality of the attack. Even if the comment line can be injected through certain methods, it does not appear seamless. For example, in a realistic scenario where a user is using Copilot to complete a code snippet line by line, the sudden appearance of a strange comment line would likely be noticeable.
> >
> > Secondly, as also mentioned in the rebuttal, the use of static analysis can easily detect the vulnerable code. This further weakens the strength of the proposed attack.
> >
> > I will maintain my current score.

---

> > > ### Comment · Reviewer_Xra8 · 2024-11-26
> > >
> > > I just wanted to address this bit:
> > >
> > > > For example, in a realistic scenario where a user is using Copilot to complete a code snippet line by line, the sudden appearance of a strange comment line would likely be noticeable
> > >
> > > I believe the authors do not suggest they insert the comment in the UI, but rather maliciously insert it into the prompt sent to the server. But I could be wrong.

---

> > > > ### Comment · Reviewer_Jqc8 · 2024-11-26
> > > >
> > > > Yes, I agree that it's insert into the prompt sent to the server. However, in a real-world scenario, the user interacts with Copilot in a recursive manner—completing one line before Copilot suggests the next. This interactive process makes it challenging to insert the comment seamlessly, as the user is likely to notice any strange comment line that appears. Unlike the prompt-response interactions typical of natural language LLMs, this iterative coding workflow leaves little room for unnoticed insertions.
> > > >
> > > > This is my understanding of how code completion works, but please correct me if I’m mistaken.

---

> ### Author Response · Authors · 2024-11-26
> **Response to Reviewer Jqc8**
>
> We thank the reviewer for engaging with our response and would like to clarify the remaining concerns. We also thank Reviewer Xra8 for participating in the discussion.
>
> ### **ReQ1: Are injected attack comments visible to the user?**
>
> No. As also pointed out by Reviewer Xra8, the malicious comment is inserted into the prompt sent to the server, which always happens at the backend and is hidden from the user interface in the frontend. Therefore, the attack comment is completely invisible to the user at all interaction steps. Moreover, the insertion of the fixed attack comment is straightforward, further increasing the feasibility of the attack.
>
> In more detail, the workflow of an interactive code completion step is as follows:
> 1) In the frontend, the user edits file buffer F.
> 2) F is sent to a black-box LLM (e.g., Copilot), which returns a completion C.
> 3) The completion C is redirected into the IDE and displayed to the user by overlaying it onto F.
>
> With INSEC applied, the workflow would look like:
> 1) In the frontend, the user edits file buffer F.
> 2) In the backend, F is copied by INSEC into a separate buffer F’.
> 3) INSEC injects its attack string into F’.
> 4) F’ is sent to a black-box LLM (e.g., Copilot), which returns a completion C’.
> 5) The completion C’ is redirected into the IDE and displayed to the user by overlaying it onto F.
>
> Note that the completions C and C’ only contain the output, e.g., the yellow part in Figure 1. They are disjoint from F and F’ respectively and do not contain the attack comment. Therefore, the attack comment is also not revealed to the user through the completion. Regardless of whether INSEC is applied or not, the user sees the same elements: the file F and the output completion. The only difference lies between C and C’. In most cases, C and C’ have the same functionality, as demonstrated in our HumanEval experiments. In security-sensitive scenarios, C’ is more likely to be insecure than C. This is the basis for our claim to INSEC’s stealthiness.
> ### **ReQ2: Can static analysis easily detect the generated vulnerable code in the wild?**
>
> No. Static analysis is often inadequate in detecting vulnerabilities in the wild [2,4,5,7]. One reason is that it is usually unknown if and which vulnerabilities a piece of code contains, preventing the choice and application of appropriate queries as done in our eval  [4,5]. Meanwhile, attempting to detect all potential vulnerabilities is limited by the CWE coverage of static analysis tools, as was already pointed out by the reviewer. Moreover, static analysis tools are found to fail at effectively explaining discovered vulnerabilities to developers [1] and lack actionable recommendations for mitigation [2]. This results in static analysis being less prevalent in practical settings than one might expect [3], with vulnerabilities generated by Copilot having already found their way into public code repositories [6]. We demonstrate that INSEC is effective on various known CWEs, which, combined with the aforementioned reasons, poses a major threat to future code security. It is also well conceivable for an attacker to adapt INSEC to target on vulnerabilities that are even more difficult to detect, novel or otherwise not covered by deployed static analyzers.
>
> It should be noted that the above argument does not contradict our use of CodeQL for security evaluation. This is because our evaluation setting is more controlled than in-the-wild vulnerability detection, in that we know how vulnerabilities can manifest in the generated code for all test cases. This allows us to carefully select a specialized CodeQL query that works effectively for each test case, as described in Section 5.1.
>
> **References**
> [1] Nachtigall et. al. "Explaining static analysis-a perspective." IEEE ASEW, 2019
> [2] Nachtigall et. al. "Evaluation of Usability Criteria Addressed by Static Analysis Tools on a Large Scale." 2023
> [3] Ryan et. al. "Unhelpful Assumptions in Software Security Research." CCS 2023
> [4] Charoenwet et. al. “An Empirical Study of Static Analysis Tools for Secure Code Review.” arXiv preprint
> [5] Mendonça et. al. "An empirical investigation on the challenges of creating custom static analysis rules for defect localization." Software Quality Journal 2022
> [6] Nguyen et. al. “Security Weaknesses of Copilot Generated Code in GitHub.” arXiv preprint
> [7] Vasallo et. al. “Context is king: The developer perspective on the usage of static analysis tools.” IEEE SANER 2018

---

> > ### Comment · Reviewer_Jqc8 · 2024-11-26
> >
> > Thank you for the clarificaiton. I increased the score accordingly.

---

> > > ### Author Response · Authors · 2024-11-27
> > > **Response to Reviewer Jqc8**
> > >
> > > We thank Reviewer Jcq8 for the insightful discussion and for their positive re-evaluation of our work. We believe that the reviewer has helped us make our paper better and will incorporate the discussed points into the next revision. With this, we will include a figure to visualize the code completion workflow with and without an attack through INSEC, as well as an adapted discussion on the use of static analysis for evaluation and defenses.
> > >
> > > We believe to have addressed all concerns raised by the reviewer in the discussion. Since the reviewer is still recommending borderline rejection, we kindly request further clarification on remaining concerns or specific aspects of the paper that fail to meet the reviewer's expectations. We are committed to continuing this productive discussion.

---

> > > > ### Comment · Reviewer_Jqc8 · 2024-11-27
> > > >
> > > > Thank you for providing all the experiments and clarifications. My final score is based on a comprehensive evaluation of current paper's technical soundness, threat model, and the feasibility of defending against the proposed attack. After careful consideration, I believe my assigned score is appropriate.

---

> > > > > ### Author Response · Authors · 2024-11-29
> > > > > **Response to Reviewer Jqc8**
> > > > >
> > > > > We thank the reviewer for their careful analysis of our work. We would like to kindly ask the reviewer to elaborate on their remaining issues with the paper's technical soundness, threat model, and the feasibility of defenses. This would greatly help us in improving the paper for future revisions.

---

### Official Review · Reviewer_384H · 2024-11-04

**Soundness:** 3
**Presentation:** 3
**Contribution:** 3
**Rating:** 6
**Confidence:** 4

**Summary:**

This paper proposes INSEC, a black-box adversarial attack designed to trick LLMs to generate vulnerable code. Evaluation on 5 programming languages and 16 CWEs show that INSEC can increase the rate of generating insecure code by ~50% while maintaining high functional correctness.

**Strengths:**

- INSEC is evaluated on a range of programming languages and CWEs, showing a high rate of vulnerability generation and maintaining close-to-original functional correctness.
- The paper presents some interesting findings. For example, better completion engines (bigger LLMs) retain more functional correctness under the attack.
- Detailed ablation study on each component of INSEC.

**Weaknesses:**

- **Dataset**: the reference solutions are generated by GPT-4 for languages other than Python.
  - There are existing extension to HumanEval, such as HumanEval-X [1] that includes human-crafted data samples and solutions in Python, C++, Java, JavaScript, and Go.
- The vulnerability metric is not rigorous enough, and thus may not truly reflect the attack success rate. CodeQL could make mistakes in vulnerability judgment (line 318-319) as prior work highlights the low precision and high false alarm rates common in static analysis tools [2].
  - Some verification is needed to confirm the vulnerability. For example, combine static analysis with dynamic testing or manual code review for a subset of samples. Alternatively, a combination of several vulnerability detection methods can be applied to lower false positives. Discuss the potential impact of false positives on the results.
- Additional evaluation could enhance the analysis, including testing out-of-distribution code samples (see Questions) and assessing robustness across various prompts.
  - For example, if the user specify "generating secure code" in the prompt, would the vulnerability ratio decrease?
- Presentation and readability could be improved
  - Define CWE upon its first use (Line 89).
  - The problem set-up paragraphs (Section 1 - 3) are lengthy and contain some redundant information (e.g., lines 68-70 and 167-169).
  - As INSEC only modifies the prefix for best performance, it should be made clear in the threat model description (line 159-160): 'INSEC modifies only the prefix p, leaving the suffix s unchanged.'
  - Consider incorporating the details and breakdown of the five programming languages and 16 CWEs into the main text for clarity.

[1] Qinkai Zheng et al. Codegeex: A pre-trained model for code generation with multilingual benchmarking on humaneval-x. In Proceedings of the 29th ACM SIGKDD Conference on Knowledge Discovery and Data Mining, KDD ’23, pp. 5673–5684, New York, NY, USA, 2023.

[2] Hong Jin Kang, Khai Loong Aw, and David Lo. Detecting false alarms from automatic static analysis tools: How far are we? In Proceedings of the 44th International Conference on Software Engineering, pp. 698–709, 2022.

**Questions:**

1. The attack strings pool and the exact attack string to inject are highly dependent on the training and validation sets (historical data). Would the attack success rate drop significantly if the test code were out-of-distribution, such as when using new API calls? Could randomization, rather than selecting the top string from the validation set, improve generalizability?
2. **Design Choices:** How were the following values chosen, and could different values impact the attack success rate?
   - "During evaluation, we generate 40 completions for each task and compute func rate@1 and func rate@10" (Line 302)
   - "generate 100 completion samples for each task" (Line 316)
3. Does the performance on vulnerability generation and functional correctness vary across different programming languages and CWEs?

---

> ### Author Response · Authors · 2024-11-25
> **Response to Reviewer 384H (Part 1)**
>
> We would like to thank the reviewer for their insightful review and constructive comments. We address their comments, questions, and concerns below:
>
> ### **Q1: Are the functional correctness (FC) results presented in the paper representative of results on other FC datasets, such as HumanEval-X?**
>
> Yes. We thank the reviewer for the pointer to HumanEval-X. It is important to note that HumanEval-X does not contain a translation for Ruby. We preferred to treat the different languages equally and thus translated all languages using GPT-4. To address the reviewers' concerns, we evaluate the functional correctness rate of INSEC on the C++ and JavaScript versions of HumanEval-X, since the Python version is the same as the one used by us. We compared these results with those in our original submission, i.e., obtained using GPT-4-generated reference solutions. The comparison of the obtained fun_rates is displayed in the table below, showing that *the numbers are similar between manually translated and GPT4-generated reference solutions*. After discussing the results on HumanEval-X with the reviewer and they still consider it relevant, we will be ready to report them in place of our translations in the next revision of the paper.
>
> | Model | HumanEval-X@1 | HumanEval-X@10 | GPT-Translated@1 | GPT-Translated@10 |
> |---|---|---|---|---|
> | StarCoder-3B | 78.0 | 99.8 | 74.9 | 97.9 |
> | CodeLlama-7B | 88.2 | 99.8 | 87.7 | 99.7 |
> | StarCoder2-3B | 89.5 | 100.2 | 90.3 | 99.5 |
> | StarCoder2-7B | 87.0 | 99.8 | 85.2 | 99.3 |
> | StarCoder2-15B | 94.6 | 100.5 | 96.1 | 100.1 |
>
>
>
> ### **Q2: Is CodeQL a suitable tool for the security evaluation presented in this paper? Can you confirm this by manual code review?**
>
> Yes. Please be aware that the traditional vulnerability detection setting differs from our evaluation setting, so the conclusion drawn from the former does not necessarily apply to the latter. In traditional vulnerability detection, it is usually unknown if a piece of code contains vulnerabilities and we agree with the reviewer that static analyzers can be inaccurate in this case. However, our evaluation setting is more controlled, in that we know how vulnerabilities can manifest in the generated code for all test cases. This allows us to carefully select a CodeQL query that works effectively for each test case as described in Section 5.1, thus achieving high accuracy for vulnerability judgment. Note that vulnerability judgment based on CodeQL is standard in the field [1,2].
>
> We also adopt the reviewer’s suggestion to manually verify the results of CodeQL, by investigating 50 generated completions of the test set from unattacked, Init-only attacked, and optimized attacked models, across all CWEs and models. The precision of CodeQL among these samples was 98%. We added the details of our manual code review and  a clarification to the paper in lines 316 and 317 and Appendix A.
>
> ### **Q3: Would additional processing of the code, such as adding comments to maintain security, invalidate the attack?**
>
> No, we investigate this by an experiment on GPT-3.5. We evaluate all CWEs, and insert the comment `This code should be secure` at the line above the INSEC attack string. We use the attack string used for the main experiments, i.e., not adaptively optimized for the presence of such a comment. While we observe that this slightly decreases the vulnerability ratio from 76% to 62%, the final ratio still largely exceeds the baseline vulnerability ratio of 22%. We added these results to the revised paper in Appendix E.
>
> Our results are not surprising, as previous work [3,4] has already shown that comments are usually not sufficient to reliably steer models towards secure generation. Further, we would like to point out that Copilot, a state-of-the-art fill-in-the-middle completion engine, claims that it applies additional sophisticated processing to prevent security vulnerabilities [5], but is successfully attacked by INSEC. We conclude that even advanced approaches are not effective in mitigating our attack.
>
> ### **Q4: How would the attack behave on code snippets different from the training and validation snippets?**
>
> The CWEs that we consider all share a similar structure in the imminent source of the bug, such as a call to a sanitizer or off-by-one memory allocation. However, the surrounding context of these locations varies strongly between various realistic settings within training, validation, and test samples. As we aim to attack only those situations, and since the attack appears to have little effect on non-critical code as detected by func_rate, we believe our attack to work in new contexts as well.
>
> Overall, it is hard to determine what would count as out-of-distribution here, since we attack specific CWEs and specific coding situations. If the reviewer has concrete suggestions of out-of-distribution data points for evaluation we would be glad to run our attack against them.

---

> > ### Author Response · Authors · 2024-11-25
> > **Response to Reviewer 384H (Part 2)**
> >
> > ###  **Q5: How were the evaluation protocol parameters chosen?**
> >
> > We list justifications for our evaluation protocol below:
> >
> > *40 completions for each task for func_rate@1/10*: We follow previous work to choose an n > k for evaluation pass@k to determine an unbiased estimator [6], with 40 much larger than 10 allowing a robust estimation.
> >
> > *100 completions for each task of vulnerability*: We choose a high, feasible value for completions to obtain a realistic estimation of how likely the model is to inject a vulnerability. Since the attack can be deployed at scale, an increase in probability is sufficient compared to a guaranteed vulnerability in one-shot scenarios.
> > We thank the reviewer for pointing out these missing details and they were added to the revised paper at Line 303.
> >
> > ###  **Q6: Does the performance on vulnerability generation and functional correctness vary across different programming languages and CWEs?**
> >
> > We thank the reviewer for this interesting question and refer to Appendix C and Figure 10 of our paper, where we already extensively discuss the performance per CWE and programming language. As the reviewer correctly suspects, the performance of vulnerability generation and functional correctness vary between programming languages and CWEs. It can be seen here that for example CWE-131, where the buffer size should be incorrectly computed, and CWE-943, data query injection, appear to be hard to attack. For functional correctness, CWE 476 and 077 (in C++ and Ruby), appear to be more heavily affected. For all other cases, the attack is highly effective.
> >
> > ### **Q7: Suggestions for improving the presentation**
> >
> > We thank the reviewer for their suggestions, they help improve our paper. We have adjusted the corresponding sections.
> >
> > Regarding the objection to the generality of the threat model: The current formulation more accurately captures the general setting of prompt injection. In Section 5.3, we also demonstrate ablations on the exact position of the attack string, including positions in the suffix. The final variant of INSEC only modifies the prefix, which we made more clear in Section 4.
> >
> > **References**
> >
> >
> > [1] Hammond Pearce et. al. “Asleep at the Keyboard? Assessing the Security of Github Copilot’s Code Contributions”. IEEE S&P, 2022
> > [2] Jingxuan He and Martin Vechev. “Large language models for code: Security hardening and adversarial testing”. CCS, 2023
> > [3] Liu, S., et. al. “From Solitary Directives to Interactive Encouragement! LLM Secure Code Generation by Natural Language Prompting”. arXiv preprint
> > [4] He, Jingxuan, et al. "Instruction tuning for secure code generation." ICML 2024
> > [5] https://github.blog/ai-and-ml/github-copilot/github-copilot-now-has-a-better-ai-model-and-new-capabilities/#filtering-out-security-vulnerabilities-with-a-new-ai-system
> > [6] Chen, Mark, et al. "Evaluating large language models trained on code." arXiv preprint arXiv preprint

---

> > > ### Comment · Reviewer_384H · 2024-11-30
> > >
> > > Thanks to the authors for the clarification and additional experiments. Most of my questions and concerns have been resolved from the response and revised version.  I am increasing my score accordingly.

---

> > > > ### Author Response · Authors · 2024-12-02
> > > >
> > > > We thank the reviewer for the insightful discussion and for raising their score towards acceptance of the paper. If the reviewer has any remaining concerns or questions, we are eager to continue the discussion.

---

### Official Review · Reviewer_Xra8 · 2024-11-04

**Soundness:** 3
**Presentation:** 3
**Contribution:** 3
**Rating:** 6
**Confidence:** 4

**Summary:**

The paper proposes a practical attack on code completion tools like Github Copilot. The authors insist (rightly so) that other attacks on code completion tools assume access to the training dataset, which is rather complicated to get, instead the paper treats these models as black boxes, and assumes access to the API (e.g., via malicious IDE extensions). The key idea is to simply insert a comment in the prompt, throwing the model off and getting it to generate vulnerable code. The paper describes a method to come up with such comments and extensively evaluates their method on different code completion tools/models. For evaluation, the authors come up with their own datasets inspired by HumanEval in order to measure both correctness and vulnerability (detected using CodeQL). Furthermore, authors perform a good number of ablations to evaluate the effectiveness of their design/heuristic choices.

**Strengths:**

1. The paper proposes a practically realizable attack scenario on code completion engines.
2. The proposed method for generating attacks is very cheap and simple to replicate.
3. Authors evaluate their proposal fairly well by performing several ablations.
4. Authors come up with a new dataset, which will be useful for the research community (when open sourced)

**Weaknesses:**

1. The authors only evaluate on their own dataset. Since the problem of getting code LMs to output vulnerable code is a well studied problem, surely there are existing datasets that can complement this evaluation. The dataset prepared here is based on HumanEval and therefore is not necessarily aligned with real coding scenarios (e.g., HumanEval problems are short). Whereas in reality, users work with one or more files and the extension uses RAG to stuff context from multiple places into the prompt. The authors can consider basing their evaluations based on other benchmarks that are closer to the real-world setting of code completion, such as repocoder (https://arxiv.org/abs/2303.12570) and repobench (https://arxiv.org/abs/2306.03091). While that would definitely make functional-correctness evaluation challenging, it would significantly strengthen the paper.

**Questions:**

N/A

---

> ### Author Response · Authors · 2024-11-25
> **Response to Reviewer Xra8**
>
> We would like to thank the reviewer for their insightful review and constructive comments. We address their comments, questions, and concerns below:
>
> ### **Q1: Can you evaluate the functional correctness on further coding benchmarks?**
>
>
> Yes, we adapt the repository-level code completion benchmark RepoBench [1] to our setting and report the achieved score below. RepoBench is based on recently created GitHub repositories to avoid contamination, contains complete files and reports Exact Match (EM), Edit Similarity (ES), and CodeBLEU (CB) scores based on a golden next line to be completed. Overall we observe only a slight performance decrease due to the attack, *matching our observations from HumanEval*.
>
> | Model                       | Unattacked   |   EM |   ES |   CB | Attacked   |   EM |   ES |   CB |
> |-----------------------------|--------------|------|------|------|------------|------|------|------|
> | StarCoder-3B            |              | 21.6 | 62.3 | 30.0 |            | 20.6 | 54.2 | 28.6 |
> | CodeLlama-7B          |              | 20.8 | 53.9 | 27.8 |            | 21.3 | 52.0 | 29.2 |
> | GPT-3.5-Turbo-Instruct |              | 29.4 | 68.3 | 36.3 |            | 25.8 | 65.7 | 33.7 |
> | StarCoder2-3B               |              | 25.4 | 66.5 | 33.9 |            | 22.3 | 55.7 | 29.7 |
> | StarCoder2-7B             |              | 26.2 | 66.1 | 33.5 |            | 23.8 | 57.9 | 31.5 |
> | StarCoder2-15B              |              | 29.0 | 66.7 | 35.5 |            | 24.7 | 58.1 | 32.5 |
>
> Concretely, we choose the first 333 samples from the Python Cross-File-First, Cross-File-Random, and In-File settings and sample 40 times completions from Starcoder 3b, the StarCoder2 family, CodeLlama 7b and GPT-3.5-Turbo-Instruct. We sample completions based on the prefix once without an attack string for the baseline setting (“Unattacked”), once with the attack string for each Python CWE, and report the average of the results in the “Attacked” setting. We will incorporate a complete investigation in the next revision of the paper.
>
> **References**
> [1] Liu, Tianyang, Canwen Xu, and Julian McAuley. "Repobench: Benchmarking repository-level code auto-completion systems." ICLR 2024

---

> ### Author Response · Authors · 2024-12-02
>
> We thank the reviewer once again for their overall positive assessment of our paper and for their engagement in the discussion with another reviewer. In case the reviewer has any remaining questions or concerns about our rebuttal or the paper, we are eager to engage in a discussion until the end of the discussion period on Tuesday this week.

---

### Author Response · Authors · 2024-11-25
**General Response to Paper Reviews**

We would like to thank the reviewers for their insightful and constructive comments. Overall we are glad that reviewers appreciated the scope of our evaluation and ablations (Ue9Z, Jqc8, 384H, Xra8), the effectiveness of our method (384H, Jqc8), and the novelty of our work (Ue9Z, Xra8).

To clarify the reviewers’ remaining questions, comments, and concerns, we have responded in individual comments.

Further, we uploaded a revised version of our paper, incorporating the requested changes and our clarifications, with changes highlighted in blue color. Overall, we made the following changes to the paper:

- Added Discussion of possible defenses (Line 485-486, Appendix E)
- Discussion of Yang et al. [1] in Related Work (Line 523-527)
- Experimental Results regarding generalization of INSEC between models and the effect of the choice of mutation function (Appendix C)
- Details regarding the manual confirmation of CodeQLs accuracy (Line 313, Appendix A)
- Introduction of prior work, which used genetic algorithms on LLM inputs (Line 197-198)
- Details regarding the choice of evaluation parameters (Line 303) and the construction of the vulnerability dataset (Line 311-312)
- Clarification that random initialization is used in the “Opt only” ablation (Line 398-401)
- Explicit introduction of Common Weakness Enumeration (Line 88)
- Several rephrasings to enhance clarity and reduce wordiness (Line 68-70, 84, 168-170, 186)


[1] Z Yang et al., Natural Attack for Pre-trained Models of Code. ICSE 2022.

---

### Meta-Review · Area_Chair_74hX · 2024-12-15

**Metareview:**

The paper proposes an attack against code models. However, the primary concerns revolve around the threat model, as well as the tools and datasets used for evaluation. These issues make the attack less convincing in real-world scenarios. As a security paper, the AC recommends that the authors revisit the threat model and present more compelling results by addressing the problem within a real-world context.

**Additional Comments On Reviewer Discussion:**

The reviewer raised concerns about the evaluation, noting that it does not necessarily align with real-world coding scenarios. Two datasets, RepoCoder and RepoBench, were suggested. However, there was limited discussion about RepoCoder, leaving the concern unresolved (although the reviewer who initially raised this issue did not reiterate it after the additional evaluation was provided).

The reviewer also questioned the use of CodeQL. The authors acknowledged that CodeQL is not an ideal choice, which further undermines the paper's persuasiveness.

Additionally, the reviewer found the practicality of the attack unconvincing. The threat model presented lacks robustness, making the proposed method appear less practical.

---

### Decision · Program_Chairs · 2025-01-22

Reject